# A First-Order Algorithmic Framework for Wasserstein Distributionally Robust Logistic Regression

**Jiajin Li, Sen Huang, Anthony Man-Cho So**
Department of Systems Engineering & Engineering Management
The Chinese University of Hong Kong
Shatin, N. T., Hong Kong
{jjli,hsen,manchoso}@se.cuhk.edu.hk

## Abstract

Wasserstein distance-based distributionally robust optimization (DRO) has received much attention lately due to its ability to provide a robustness interpretation of various learning models. Moreover, many of the DRO problems that arise in the learning context admits exact convex reformulations and hence can be tackled by off-the-shelf solvers. Nevertheless, the use of such solvers severely limits the applicability of DRO in large-scale learning problems, as they often rely on general purpose interior-point algorithms. On the other hand, there are very few works that attempt to develop fast iterative methods to solve these DRO problems, which typically possess complicated structures. In this paper, we take a first step towards resolving the above difficulty by developing a first-order algorithmic framework for tackling a class of Wasserstein distance-based distributionally robust logistic regression (DRLR) problem. Specifically, we propose a novel linearized proximal ADMM to solve the DRLR problem, whose objective is convex but consists of a smooth term plus two non-separable non-smooth terms. We prove that our method enjoys a sublinear convergence rate. Furthermore, we conduct three different experiments to show its superb performance on both synthetic and real-world datasets. In particular, our method can achieve the same accuracy up to 800+ times faster than the standard off-the-shelf solver.

## 1   Introduction

One of the basic principles for dealing with the overfitting phenomenon in statistical learning is regularization [23]. Recently, there has been a flurry of works that aim to interpret regularization from a distributionally robust optimization (DRO) perspective; see, e.g., [19, 1, 7, 20, 18] and the references therein. The results in these works not only provide a probabilistic justification of existing regularization techniques but also offer a powerful alternative approach to tackle risk minimization problems. Indeed, it has been shown that the DRO formulations of various statistical learning problems admit polynomial-time solvable and exact convex reformulations [19, 7, 2, 21, 20], which can be tackled by off-the-shelf solvers (e.g., YALMIP). Nevertheless, the use of such solvers severely limits the applicability of the DRO approach in large-scale learning problems, as they often rely on general-purpose interior-point algorithms. On the other hand, there are very few works that address the design of fast iterative methods for solving the convex reformulations of DRO problems. This is in part due to the complicated structures that are often possessed by such reformulations. In fact, it is only recently that researchers have proposed stochastic gradient descent (SGD) algorithms for DRO with $f$-divergence-based ambiguity sets [17]. However, $f$-divergence measures can only compare distributions with the same support, while the Wasserstein distances do not have such a restriction. On another front, the works [15, 11] propose cutting-surface methods to deal with Wasserstein distance-based DRO problems. However, they tend to suffer a large computational burden.

In this paper, we take a first step towards bridging the above-mentioned gap by proposing a new first-order algorithmic framework for solving the class of Wasserstein distance-based distributionally robust logistic regression (DRLR) problems considered in [19]. The starting point of our investigation is the following reformulation result; see Theorem 1 and Remark 2 in [19]:

$$\inf_{\beta} \sup_{\mathbb{Q} \in B_\epsilon(\hat{\mathbb{P}}_N)} \mathbb{E}_{(x,y)\sim\mathbb{Q}}[\ell_\beta(x,y)] \triangleq \inf_{\beta,\lambda} \lambda\epsilon + \frac{1}{N}\sum_{i=1}^{N}\left(\ell_\beta(\hat{x}_i,\hat{y}_i) + \max\{\hat{y}_i\beta^T\hat{x}_i - \lambda\kappa, 0\}\right) \tag{1.1}$$
$$\text{s.t. } \|\beta\|_* \le \lambda.$$

Here, $x \in \mathbb{R}^n$ denotes a feature vector and $y \in \{-1, +1\}$ its associated label to be predicted; $\ell_\beta(x,y) = \log(1 + \exp(-y\beta^Tx))$ is the log-loss associated with the feature-label pair $(x,y)$ and regression parameter $\beta \in \mathbb{R}^n$; $\{(\hat{x}_i,\hat{y}_i)\}_{i=1}^N$ are $N$ training samples drawn from an unknown underlying distribution $\mathbb{P}^*$ on the feature-label space $\Theta = \mathbb{R}^n \times \{-1, +1\}$; $\hat{\mathbb{P}}_N = \frac{1}{N}\sum_{i=1}^{N}\delta_{(\hat{x}_i,\hat{y}_i)}$ denotes the empirical distribution associated with the training samples $\{(\hat{x}_i,\hat{y}_i)\}_{i=1}^N$; $B_\epsilon(\hat{\mathbb{P}}_N) = \{\mathbb{Q} \in \mathcal{P}(\Theta) : W(\mathbb{Q}, \hat{\mathbb{P}}_N) \le \epsilon\}$ is the ball in the space $\mathcal{P}(\Theta)$ of probability distributions on $\Theta$ that is centered at the empirical distribution $\hat{\mathbb{P}}_N$ and has radius $\epsilon$ with respect to the Wasserstein distance

$$W(\mathbb{Q}, \hat{\mathbb{P}}_N) = \inf_{\Pi \in \mathcal{P}(\Theta \times \Theta)}\left\{\int_{\Theta \times \Theta}d(\xi, \xi')\Pi(\mathrm{d}\xi, \mathrm{d}\xi') : \Pi(\mathrm{d}\xi, \Theta) = \mathbb{Q}(\mathrm{d}\xi), \Pi(\Theta, \mathrm{d}\xi') = \hat{\mathbb{P}}_N(\mathrm{d}\xi')\right\},$$

where $\xi = (x,y) \in \Theta$, $d(\xi, \xi') = \|x - x'\| + \frac{\kappa}{2}|y - y'|$ is the transport cost between two data points $\xi, \xi' \in \Theta$ induced by a generic norm $\|\cdot\|$ on $\mathbb{R}^n$ with $\|\cdot\|_*$ being its dual norm, and $\kappa > 0$ is a parameter that represents the reliability of the label measurements (the larger the $\kappa$, the more reliable are the measurements; when $\kappa = \infty$, the measurements are error-free). The formulation on the left-hand side of (1.1) is motivated by the desire to construct an ambiguity set around the empirical distribution $\hat{\mathbb{P}}_N$ that contains the true distribution $\mathbb{P}^*$, so that the resulting classifier has good out-of-sample performance. We refer the reader to [19] for a more detailed discussion.

A natural question that arises from (1.1) is how to solve the convex optimization problem on the right-hand side (RHS) efficiently. When $\kappa = \infty$, the RHS of (1.1) reduces to a classic regularized logistic regression problem [19, Remark 1]. As such, a host of practically efficient first-order methods (such as proximal gradient-type methods or stochastic (variance-reduced) gradient methods) with provable convergence guarantees (see, e.g., [22, 24, 28, 2]) can be applied. However, the algorithmic aspects of the practically more relevant case where $\kappa < \infty$ have not been well explored. Our proposed framework for tackling this case consists of two steps. First, by considering the optimality conditions of the RHS of (1.1), we can derive an upper bound $\lambda^U$ on the optimal $\lambda^*$. This suggests that we can first initialize $\lambda$ to a value in $[0, \lambda^U]$ and solve the resulting problem that involves only the variable $\beta$ (the $\beta$-subproblem), then apply golden-section search to update $\lambda$, and then repeat the whole process until we find the optimal solution to (1.1). Second, which is the core step of our framework, is to design a fast iterative method for solving the $\beta$-subproblem. By treating $\lambda$ as a constant, the RHS of (1.1) is equivalent to

$$\inf_{\|\beta\|_* \le \lambda} \frac{1}{N}\sum_{i=1}^{N}\left(h(\hat{y}_i\beta^T\hat{x}_i) + \max\{\hat{y}_i\beta^T\hat{x}_i - \lambda\kappa, 0\}\right) \tag{1.2}$$

with $h(u) = \log(1 + \exp(-u))$. Although (1.2) has a relatively simple norm-ball constraint, its objective is non-smooth and non-separable. As such, most existing first-order methods (e.g., projected/proximal subgradient methods) are ill-suited for tackling it. To proceed, we apply the operator splitting technique to reformulate (1.2) as

$$\inf_{\beta,\mu} \frac{1}{N}\sum_{i=1}^{N}\left(h(\mu_i) + \max\{\mu_i - \lambda\kappa, 0\}\right) \tag{1.3}$$
$$\text{s.t. } Z\beta - \mu = 0, \|\beta\|_* \le \lambda,$$

where $Z$ is the $N \times n$ matrix whose $i$-th row is $\hat{y}_i\hat{x}_i^T$, and propose a new linearized proximal alternating direction method of multipliers (LP-ADMM) to fully exploit the structure of (1.3). In particular, our method differs substantially from the commonly used ADMM-variants in the literature [27, 14, 6, 12, 8, 9, 25] in the updates of the variables. For the $\beta$-update, we solve a norm-constrained

quadratic optimization problem. Since such a problem can be rather ill-conditioned, we provide three different types of solvers to handle this task, namely, the accelerated projected gradient descent, coordinate minimization [10], and active set conjugate gradient methods [5]. For the $\mu$-update, observing that the coupling matrix for $\mu$ in the linear equality constraint is the identity, the augmented Lagrangian function is already locally strongly convex in $\mu$. Hence, instead of using a quadratic approximation of $h(\cdot)$ as in the vanilla proximal ADMM, we use a first-order approximation without step size selection; i.e.,

$$\mu^{k+1} = \arg\min_\mu \left\{ \frac{1}{N} \sum_{i=1}^N \left( h'(\mu_i^k)\mu_i + \max\{\mu_i - \lambda\kappa, 0\} \right) - (w^k)^T (Z\beta^{k+1} - \mu) + \frac{\rho}{2}\|\mu - Z\beta^{k+1}\|_2^2 \right\},$$

where $w \in \mathbb{R}^N$ is the dual variable associated with the linear equality constraint in (1.3) and $\rho > 0$ is the penalty parameter in the augmented Lagrangian function. On the theoretical side, we prove that our proposed LP-ADMM enjoys an $\mathcal{O}(\frac{1}{K})$ convergence rate under standard assumptions. On the numerical side, we demonstrate via extensive experiments that our proposed method can be sped up substantially by adopting a geometrically increasing step size strategy. In particular, our method can achieve a hundred-fold speedup over the standard solver (which is the only other method that has been used so far to solve (1.1)) on both synthetic and real-world datasets without the need to tune an optimal penalty parameter in every iteration. To the best of our knowledge, our work is the first to propose a first-order algorithmic framework for solving the Wasserstein distance-based DRLR problem (1.1) for any $\kappa > 0$. Moreover, the proposed framework is sufficiently general that it can potentially be applied to other DRO problems, which could be of independent interest.

## 2    Preliminaries

Let us introduce some basic definitions and concepts. To allow for greater generality, consider the following problem:

$$\begin{aligned} \underset{x,y}{\text{minimize}} \quad & F(x,y) = f(y) + P(y) + g(x) \\ \text{subject to} \quad & Ax - y = 0. \end{aligned} \tag{2.1}$$

Here, $f : \mathbb{R}^N \to \mathbb{R}$ is a closed convex function that is continuously differentiable on $\text{int}(\text{dom}(f))$ with linear operator $A \in \mathbb{R}^{N \times n}$; $P : \mathbb{R}^N \to \mathbb{R} \cup \{+\infty\}$ is a closed proper convex function; $g(x)$ is the indicator function of a norm ball. It should be clear that problem (2.1) includes the $\beta$-subproblem (1.3) as a special case. Indeed, the latter can be written as

$$\begin{aligned} \underset{\mu,\beta}{\text{minimize}} \quad & F(\mu,\beta) = f(\mu) + P(\mu) + g(\beta) \\ \text{subject to} \quad & Z\beta - \mu = 0, \end{aligned}$$

where $f(\mu) = \frac{1}{N}\sum_{i=1}^N \left\{ \log(1 + \exp(-\mu_i)) + \frac{1}{2}(\mu_i - \lambda\kappa) \right\}$, $P(\mu) = \frac{1}{2N}\sum_{i=1}^N |\mu_i - \lambda\kappa|$, and $g(\beta) = \mathbb{I}_{\{\|\beta\|_* \le \lambda\}}$. Now, the augmented Lagrangian function associated with (2.1) is given by

$$\mathcal{L}_\rho(x,y;w) = f(y) + P(y) + g(x) - w^T(Ax - y) + \frac{\rho}{2}\|Ax - y\|_2^2, \tag{2.2}$$

where $w$ is the multiplier. We use $(\mathcal{X}^*, \mathcal{Y}^*)$ to denote the solution set of (2.1). A point $(x^*, y^*)$ is optimal for (2.1) if there exists a $w^*$ such that the following KKT conditions are satisfied:

$$\begin{cases} A^T w^* \in \partial g(x^*), \\ -w^* \in \nabla f(y^*) + \partial P(y^*), \\ Ax^* - y^* = 0. \end{cases} \tag{2.3}$$

**Assumption 2.1.** *There exists a point $(x^*, y^*, w^*)$ satisfying the KKT conditions in* (2.3).

**Assumption 2.2.** *The gradient of the function $f$ is Lipschitz continuous; i.e., there exists a constant $L_f > 0$ such that*

$$\|\nabla f(x) - \nabla f(y)\| \le L_f \|x - y\|, \forall x, y.$$

**Definition 2.3** (Bregman Divergence). *Let $f : \Omega \to \mathbb{R}$ be a function that is a) strictly convex, b) continuously differentiable, and c) defined on a closed convex set $\Omega$. The Bregman divergence with respect to $f$ is defined as*

$$B_f(x,y) = f(x) - f(y) - \langle \nabla f(y), x - y \rangle.$$

## 3 First-Order Algorithmic Framework

In this section, we present our first-order algorithmic framework for solving the DRLR problem. For concreteness' sake, we take $\|\cdot\|$ in the transport cost to be the $\ell_1$-norm in this paper. However, it should be mentioned that our framework is general enough to handle other norms as well.

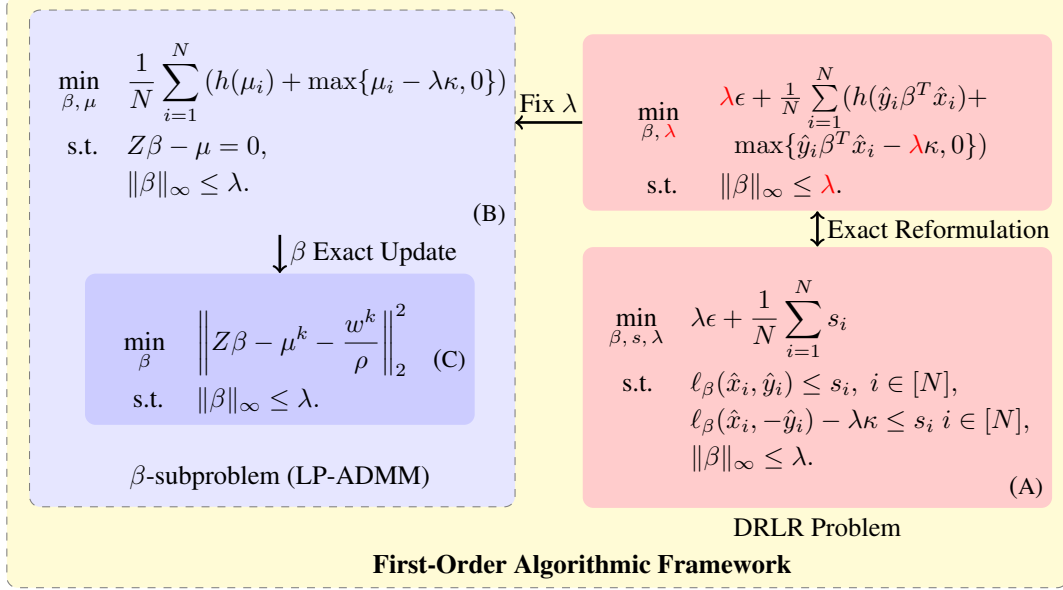

Figure 1: First-order algorithmic framework for Wasserstein DRLR with $\ell_1$-induced transport cost

We summarize the key components of our first-order algorithmic framework in Figure 1. As shown in [19], the original DRLR problem (i.e., LHS of (1.1)) can be reformulated as the convex program (A) using strong duality. A standard approach to tackling problem (A) is to use an off-the-shelf solver (e.g., YALMIP). To develop an efficient algorithmic framework, we focus on the RHS of (1.1) and proceed in two steps. Motivated by the structure of the RHS of (1.1), a natural first step is to fix $\lambda$ to a certain value to obtain the problem (1.2), which involves only the variable $\beta$ and will be referred to as the $\beta$-subproblem in the sequel. The second, which is also the core step of our framework, is to design a fast iterative algorithm to tackle the $\beta$-subproblem (1.2). The main difficulty of problem (1.2) comes from the two non-smooth non-separable terms. To overcome this difficulty, we introduce the auxiliary variable $\mu_i = \hat{y}_i \beta^T \hat{x}_i$ to split the non-separable non-smooth term $\max\{\hat{y}_i \beta^T \hat{x}_i - \lambda\kappa, 0\}$, thus leading to problem (B). Then, we propose a novel linearized proximal ADMM (LP-ADMM) algorithm to solve it efficiently. As will be shown in Section 4, the proposed LP-ADMM will converge at the rate $\mathcal{O}(1/K)$ when applied to the $\beta$-subproblem (B). In each iteration of our LP-ADMM algorithm, we perform an exact minimization for the $\beta$-update, which entails solving the box-constrained quadratic optimization problem (C) (here, $w^k$ denotes the corresponding Lagrange multiplier). Towards that end, we provide three alternative solvers for problem (C), which target three different settings. Specifically, we use accelerated projected gradient descent in the well-conditioned case; coordinate minimization [10] in the high-dimensional case $N \ll d$; active set conjugate gradient method [5] in the ill-conditioned case. The details are given in Appendix B.

To implement the above framework, let us first show that there is a finite upper bound $\lambda^U$ on the optimal $\lambda^*$ to problem (A). Observe that the objective function in the RHS of (1.1) takes the form

$$\Omega(\lambda, \beta) = \lambda\epsilon + \frac{1}{N}\sum_{i=1}^{N}\left(h(\hat{y}_i\beta^T\hat{x}_i) + \max\{\hat{y}_i\beta^T\hat{x}_i - \lambda\kappa, 0\}\right) + \mathbb{I}_{\{\|\beta\|_\infty \leq \lambda\}}.$$

Now, let $q(\lambda) = \inf_\beta \Omega(\lambda, \beta)$. As the function $\Omega(\cdot, \cdot)$ is jointly convex, we can conclude that $q(\cdot)$ is a convex (and hence unimodal) function on $\mathbb{R}$. Furthermore, the DRLR problem (A) satisfies the Mangasarian-Fromovitz constraint qualification (MFCQ), which implies that its KKT conditions are necessary and sufficient for optimality. As the following proposition shows, we can use the KKT system of problem (A) to derive the desired upper bound on $\lambda^*$:

**Proposition 3.1.** *Suppose that* $(\beta^*, \lambda^*, s^*)$ *is an optimal solution to problem* (A) *in Figure 1. Then, we have* $\lambda^* \leq \lambda^U = \frac{0.2785}{\epsilon}$.

*Proof.* Using $a_{ij}$, where $i = 1, \ldots, N$ and $j = 1, \ldots, 4$ to denote the multipliers associated with the constraints in problem (A), we can write down the KKT conditions of problem (A) as follows:

$$
\min_{\beta,\, s,\, \lambda} \lambda\epsilon + \frac{1}{N}\sum_{i=1}^{N} s_i
$$
$$
\text{s.t. } \ell_\beta(\hat{x}_i, \hat{y}_i) \leq s_i,
$$
$$
i \in [N],
$$
$$
\ell_\beta(\hat{x}_i, -\hat{y}_i) - \lambda\kappa \leq s_i,
$$
$$
i \in [N],
$$
$$
e_i^T\beta \leq \lambda,\ i \in [N],
$$
$$
-e_i^T\beta \leq \lambda,\ i \in [N]
$$
$$
\Rightarrow
\begin{cases}
\displaystyle\sum_{i=1}^{N} a_{i1}\nabla_\beta \ell_\beta(\hat{x}_i, \hat{y}_i) + a_{i2}\nabla_\beta \ell_\beta(\hat{x}_i, -\hat{y}_i) + (a_{i3} - a_{i4})e_i = 0, \\[2mm]
\displaystyle\sum_{i=1}^{N} \kappa a_{i2} + a_{i3} + a_{i4} = \epsilon, \\[2mm]
a_{i1} + a_{i2} = \dfrac{1}{N},\ i \in [N], \\[1mm]
a_{i1}(\ell_\beta(\hat{x}_i, \hat{y}_i) - s_i) = 0,\ i \in [N], \\[1mm]
a_{i2}(\ell_\beta(\hat{x}_i, -\hat{y}_i) - \lambda\kappa - s_i) = 0,\ i \in [N], \\[1mm]
a_{i3}(e_i^T\beta - \lambda) = 0,\ i \in [N], \\[1mm]
a_{i4}(e_i^T\beta + \lambda) = 0,\ i \in [N], \\[1mm]
a_{ij} \geq 0,\ i \in [N],\ j \in [4].
\end{cases}
$$

After some elementary manipulations (see Appendix A for details), we obtain

$$
\lambda \leq \frac{1}{N\epsilon}\sum_{i=1}^{N} \frac{\hat{y}_i\beta^T\hat{x}_i \exp(-\hat{y}_i\beta^T\hat{x}_i)}{1 + \exp(-\hat{y}_i\beta^T\hat{x}_i)} \leq \frac{0.2785}{\epsilon},
$$

as desired. $\qquad\square$

**Remark 3.2.** *Although Proposition 3.1 applies to the case where the transport cost is induced by the* $\ell_1$*-norm, the techniques used to prove it can also carry over to the* $\ell_2$ *and* $\ell_\infty$ *cases. All we need to do is to modify the parts highlighted in blue above. Indeed, when the transport cost is induced by the* $\ell_2$*-norm, the norm constraint in problem* (A) *becomes* $\|\beta\|_2 \leq \lambda$, *which is equivalent to* $\|\beta\|_2^2 \leq \lambda^2$. *On the other hand, when the transport cost is induced by the* $\ell_\infty$*-norm, the norm constraint becomes* $\|\beta\|_1 \leq \lambda$, *which can be expressed as* $B\beta \leq \lambda e_{2^n}$ *with* $B$ *being the* $2^n \times n$ *matrix whose rows are all the possible arrangements of* $+1$'s *and* $-1's$.

Proposition 3.1, together with the unimodality of $q(\cdot)$, suggests the following natural strategy for finding an optimal solution $(\beta^*, \lambda^*, s^*)$ to problem (A): initialize $\lambda$ in (A) to a value in $[0, \lambda^U]$, solve the resulting $\beta$-subproblem (B), apply golden-section search to update $\lambda$, and repeat. The pseudo-code for the golden-section search on $\lambda$ can be found in Appendix B. The $\beta$-subproblem (B) will be solved by our proposed LP-ADMM, which we present next.

## 4  LP-ADMM for the $\beta$-Subproblem and Its Convergence Analysis

To simplify notation, we consider the prototypical form (2.1) of the $\beta$-subproblem here. It can be shown that the $\beta$-subproblem (B) satisfies Assumptions 2.1 and 2.2 in Section 2. Now, we present our proposed LP-ADMM in Algorithm 1.

The $x$-update is standard in ADMM-type algorithms and leads to a box-constrained quadratic optimization problem. The crux of our algorithm lies in the local model used to perform the $y$-update. To understand the local model, observe that since the coupling matrix for $y$ in the constraint $Ax - y = 0$ is the identity, the augmented Lagrangian function $\mathcal{L}_\rho(\cdot, \cdot; \cdot)$ in (2.2) is strongly convex in $y$. Thus, instead of using the quadratic approximation of $f(\cdot)$ as in the vanilla proximal ADMM, we can use the first-order approximation $y \mapsto \hat{f}(y; y^k) = f(y^k) + \nabla f(y^k)^T(y - y^k)$ at the current iterate $y^k$. This leads to the $y$-update

$$
y^{k+1} = \arg\min_y \left\{ \hat{f}(y; y^k) - \langle w^k, Ax^{k+1} - y \rangle + \frac{\rho}{2}\|y - Ax^{k+1}\|_2^2 + P(y) \right\},
$$

which, as can be easily verified, is equivalent to the update given in Algorithm 1. In fact, using such a first-order local model not only makes the resulting algorithm converge faster in practice but also eliminates the need to perform step size selection. The latter makes our algorithmic framework numerically more robust in general.

**Algorithm 1:** Linearized Proximal ADMM (LP-ADMM) for Solving (2.1)

---

**Input**: Choose initial point $(x^0, y^0, w^0) \in \mathbb{R}^n \times \mathbb{R}^N \times \mathbb{R}^N$ and number of iterations $K$;
      Initialized the penalty parameter $\rho_0$ and shrinking parameter $\gamma \geq 1$;
**Output**: $\{(x^k, y^k, w^k)\}_{k=1}^K$ and $\{F(x^k, y^k)\}_{k=1}^K$;

1 **for** *each iteration* **do**
2

$$x^{k+1} = \arg\min_{x \in \mathbb{R}^n} \left\{ \frac{\rho_k}{2} \left\| Ax - y^k - \frac{w^k}{\rho_k} \right\|_2^2 + g(x) \right\};$$

$$y^{k+1} = \arg\min_{y \in \mathbb{R}^N} \left\{ \frac{\rho_k}{2} \left\| y - \left( Ax^{k+1} - \frac{w^k + \nabla f(y^k)}{\rho_k} \right) \right\|_2^2 + P(y) \right\};$$

$$w^{k+1} = w^k - \rho_k(Ax^{k+1} - y^{k+1});$$

$$\rho_{k+1} = \gamma\rho_k \text{ (in particular, if } \gamma = 1, \text{ then } \rho_k = \rho_{k+1} = \rho);$$

3 **end**

---

Next, let us analyze the convergence behavior of the LP-ADMM. Based on the definition of the augmented Lagrangian function in (2.2), the optimality conditions of the subproblems in Algorithm 1 can be written as follows:

$$0 \in \rho A^T \left( Ax^{k+1} - y^k - \frac{w^k}{\rho} \right) + \partial g(x^{k+1}), \tag{4.1}$$

$$0 \in \nabla f(y^k) + \rho \left( y^{k+1} - Ax^{k+1} + \frac{w^k}{\rho} \right) + \partial P(y^{k+1}). \tag{4.2}$$

Using (4.1) and (4.2), we can establish the following basic properties concerning the iterates of our proposed LP-ADMM. The proofs can be found in Appendix A.

**Proposition 4.1.** *Suppose that we use a constant penalty parameter $\rho$ that satisfies $\rho > (\sqrt{3}+1)L_f$. Let $\{(x^k, y^k, w^k)\}_{k \geq 0}$ be the sequence generated by the LP-ADMM and $(x^*, y^*, w^*)$ be a point satisfying the KKT conditions (2.3) with $x^* \in \mathcal{X}, y^* \in \mathcal{Y}$. Then, the following hold:*

(a) *For all $k \geq 1$, $\|Ax^{k+1} - y^k\|_2^2 \geq \frac{1}{2}\|y^{k+1} - y^k\|_2^2 - \frac{L_f^2}{\rho^2}\|y^k - y^{k-1}\|_2^2$.*

(b) *For all $k \geq 0$ and $(x, y)$ satisfying $Ax - y = 0$, we have $F(x^{k+1}, y^{k+1}) - F(x, y) \leq \frac{1}{2\rho}(\|w^k\|_2^2 - \|w^{k+1}\|_2^2) + \frac{\rho}{2}(\|y^k - y\|_2^2 - \|y^{k+1} - y\|_2^2) + c(\|y^k - y^{k-1}\|_2^2 - \|y^{k+1} - y^k\|_2^2) + (B_f(y, y^{k+1}) - B_f(y, y^k))$, where $c = \frac{\rho - 2L_f}{4}$.*

(c) *The sequence $\{\frac{1}{2\rho}\|w^k - w^*\|_2^2 + \frac{\rho}{2}\|y^k - y^*\|_2^2 - B_f(y^*, y^k)\}_{k \geq 0}$ is non-increasing and bounded below.*

Armed with Proposition 4.1, we can prove the main convergence theorem for LP-ADMM.

**Theorem 4.2.** *Consider the setting of Proposition 4.1. Set $\bar{x}^K = \frac{1}{K}\sum_{k=1}^K x^k$ and $\bar{y}^K = \frac{1}{K}\sum_{k=1}^K y^k$. Then, the following hold:*

(a) *The sequence $\{(x^k, y^k, w^k)\}_{k \geq 0}$ converges to a KKT point of problem (2.1).*

(b) *The sequence of function values converges at the rate $\mathcal{O}(\frac{1}{K})$:*

$$F(\bar{x}^K, \bar{y}^K) - F(x^*, y^*) \leq \frac{(1/2\rho)\|w^0\|_2^2 + (\rho/2)\|y^* - y^0\|_2^2 + c\|y^0 - y^1\|_2^2}{K} = \mathcal{O}(\frac{1}{K}).$$

**Remark 4.3.** *The standard linearized ADMM in [14, 26, 25] involves the quadratic term $\frac{\eta}{2}\|y - y^k\|^2$, where $\eta$ needs to satisfy $\eta > L_f$. Our LP-ADMM can be regarded as a linearized ADMM with $\eta = 0$. Using the first-order local model, the LP-ADMM achieves the fastest single-step update. Moreover, it is worth noting that the adaptive penalty strategy works well in practice, especially the geometrical increasing one (i.e., the blue line in Algorithm 1).*

# 5 Experiment Results

In this section, we present numerical results to demonstrate the effectiveness and efficiency of the different components in our proposed algorithmic framework. All experiments were conducted using MATLAB R2018a on a computer running Windows 10 with Intel® Core™ i5-8600 CPU (3.10 GHz) and 16 GB RAM. We conducted three different experiments to validate our theoretical results and show the high efficiency of our implementation of the proposed first-order algorithmic framework. To begin, we compare the CPU time of our framework with the YALMIP solver used by [19] on both synthetic and real datasets. Then, we present an empirical comparison of our LP-ADMM with other baseline first-order algorithms, including Projected SubGradient Method (SubGradient), Primal-Dual Hybrid Gradient (PDHG), Linearized-ADMM and Standard ADMM, on the $\beta$-subproblem. Lastly, we show the test data performance of the DRLR model on real datasets. We use the active set conjugate gradient method to solve the box-constrained quadratic optimization problem (C) in this section. Our code is available at `https://github.com/gerrili1996/DRLR_NIPS2019_exp`.

## 5.1 CPU Time Comparison with the YALMIP Solver

Our setup for the synthetic experiments is as follows. We first generate $\beta$ from the standard $n$-dimensional Gaussian distribution $\mathcal{N}(0, I_n)$ and normalize it to obtain the ground truth $\beta_* = \beta/\|\beta\|$. Next, we generate the feature vectors $\{\hat{x}_i\}_{i=1}^N$ independently and identically (i.i.d) from $\mathcal{N}(0, I_n)$ and the noisy measurements $\{z_i\}_{i=1}^N$ i.i.d from the uniform distribution over $[0, 1]$. Lastly, we compute the ground truth labels $\{\hat{y}_i\}_{i=1}^N$ via $\hat{y}_i = 2 \times \text{int}(z_i < \frac{1}{1+\exp(-\beta_*^T \hat{x}_i)}) - 1$. We set the DRLR model parameters to be $\kappa = 1, \epsilon = 0.1$ and the default parameters of our Adaptive LP-ADMM to be $\rho_0 = 0.001, \gamma = 1.05$. All the experiment results reported here were averaged over 30 independent trials over random seeds. Table 1 summarizes the comparison of CPU times on different scales in the synthetic setting. Our experiment results indicate that the proposed LP-ADMM with adaptive penalty strategy can be over 800 times faster than YALMIP, a state-of-the-art optimization solver, and the performance gap grows considerably with problem size.

Table 1: CPU time comparison: LP-ADMM vs. YALMIP (used in [19]) in the synthetic setting

| $(N, d)$ | YALMIP (s) | Non-Adaptive (s) | Adaptive (s) | Ratio |
|---|---|---|---|---|
| (10,3) | $2.40 \pm 0.18$ | $0.06 \pm 0.02$ | $0.07 \pm 0.02$ | **37** |
| (100,3) | $3.29 \pm 0.05$ | $0.14 \pm 0.03$ | $0.06 \pm 0.01$ | **54** |
| (100,10) | $3.34 \pm 0.03$ | $0.21 \pm 0.03$ | $0.08 \pm 0.01$ | **44** |
| (500,10) | $7.92 \pm 0.17$ | $0.58 \pm 0.16$ | $0.14 \pm 0.01$ | **55** |
| (500,50) | $8.53 \pm 0.17$ | $0.60 \pm 0.03$ | $0.24 \pm 0.01$ | **36** |
| (1000,50) | $16.44 \pm 0.44$ | $0.96 \pm 0.07$ | $0.25 \pm 0.02$ | **67** |
| (1000,100) | $19.16 \pm 0.48$ | $1.69 \pm 0.11$ | $0.38 \pm 0.01$ | **51** |
| (3000,50) | $65.87 \pm 1.54$ | $2.40 \pm 0.15$ | $0.32 \pm 0.01$ | **206** |
| (3000,100) | $113.94 \pm 2.05$ | $3.84 \pm 0.20$ | $0.47 \pm 0.04$ | **243** |
| (5000,100) | $287.67 \pm 2.67$ | $7.08 \pm 0.66$ | $0.64 \pm 0.03$ | **451** |
| (10000,10) | $283.25 \pm 18.98$ | $5.03 \pm 0.76$ | $0.50 \pm 0.02$ | **563** |
| (10000,100) | $1165.40 \pm 26.52$ | $19.75 \pm 3.74$ | $1.37 \pm 0.12$ | **852** |

We also tested our proposed method on the real datasets `a1a-a9a` downloaded from LIBSVM[1]. Note that the data matrices from these datasets are ill-conditioned and highly sparse, which should be contrasted with the well-conditioned and dense ones in the synthetic setting. Table 2 shows the comparison of CPU times on the real datasets. We observe that our methods work exceptionally well, especially in the large-scale case (i.e., `a9a`).

As standard ADMM-type algorithms are very sensitive to the choice of penalty parameters, it is hard for us to tune the optimal penalty parameter for each subproblem with different $\lambda$. Thus, we use a constant penalty parameter for the non-adaptive case. In fact, since we do not need to perform a careful penalty parameter selection in our method, we can achieve an even greater speed by using an adaptive penalty strategy. Moreover, it is worth noticing that our approaches achieve higher-accuracy solutions compared with the YALMIP solver in all the experiments.

Table 2: CPU time comparison: LP-ADMM vs. YALMIP (used in [19]) on UCI adult datasets

| Dataset | Data statistics | | CPU time (s) | | Ratio |
|---------|---------|----------|--------|-------|-------|
| | Samples | Features | YALMIP | Ours | |
| a1a | 1605 | 123 | 25.63 | **2.93** | **9** |
| a2a | 2265 | 123 | 39.20 | **3.53** | **11** |
| a3a | 3185 | 123 | 57.79 | **4.26** | **14** |
| a4a | 4781 | 123 | 105.32 | **4.56** | **23** |
| a5a | 6414 | 123 | 155.42 | **4.39** | **35** |
| a6a | 11220 | 123 | 413.65 | **4.68** | **88** |
| a7a | 16100 | 123 | 738.12 | **5.41** | **137** |
| a8a | 22696 | 123 | 1396.45 | **5.81** | **240** |
| a9a | 32561 | 123 | 2993.30 | **7.08** | **423** |

Figure 2: CPU time comparison with YALMIP using interior point algorithm

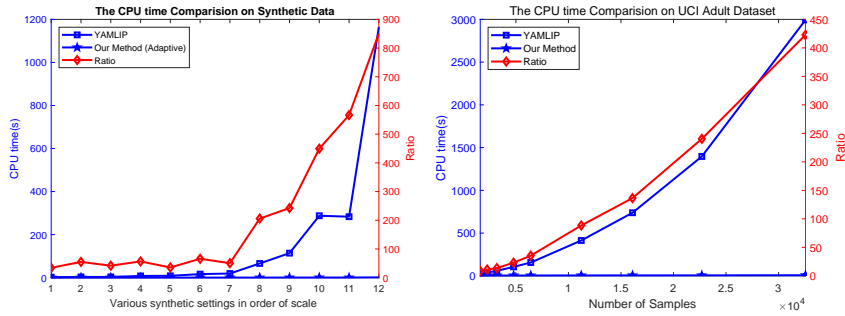

## 5.2 Efficiency of LP-ADMM for $\beta$-Subproblem

To further demonstrate the efficacy of our proposed LP-ADMM on the $\beta$-subproblem, we present an empirical comparison of our algorithm with other first-order methods in the synthetic setting. The implementation details are given as follows: $\lambda$ is regarded as a constant (i.e., $\lambda = 0.1$), the DRLR model parameters are the same as in Section 5.1, and the first-order methods used include

(a) Two-block Standard ADMM (SADMM) [3]: for both $\beta$- and $\mu$-updates, we perform exact minimization, which are done using accelerated projected gradient descent and semi-smooth Newton method [13], respectively (pseudo-codes are given in Appendix B);

(b) Primal-Dual Hybrid Gradient (PDHG) [4];

(c) Linearized-ADMM (LADMM): all the ingredients are the same as LP-ADMM, except that the $\mu$-update involves the classic quadratic term;

(d) Projected Subgradient Method (SubGradient).

The convergence curves for various synthetic cases are shown in Figure 3. The performance of our methods significantly dominates those of other methods, which agrees with our theoretical findings in Section 4. Compared with LADMM, we show practical advantages of the first-order local model for the $\mu$-update. In addition, LP-ADMM and Adaptive LP-ADMM have similar performance in small instances, but the latter has better performance in large instances. In summary, we have demonstrated that the usefulness and efficiency of all the components in our first-order algorithmic framework. In particular, as the data matrices in the real datasets are ill-conditioned, all the baseline approaches cannot achieve a high accuracy but our proposed LP-ADMM can.

## 5.3 Test Data Performance of the DRLR Model

In this subsection, we compare the test data performance of the DRLR model with two classic models, namely, Logistic Regression (LR) and Regularized Logistic Regression (RLR). The latter refers to $\min \frac{1}{N} \sum_{i=1}^{N} \ell_\beta(\hat{x}_i, \hat{y}_i) + \epsilon\|\beta\|_\infty$. If the training data labels are error-free (which corresponds to $\kappa = \infty$), then the DRLR model reduces to RLR [19]. We use grid search with cross-validation to

Figure 3: Comparison of LP-ADMM with other first-order methods on $\beta$-subproblem: $y$-axis is the sub-optimality gap $\log(F^k - F^*)$; "total iterations" refers to that taken by Adaptive ADMM

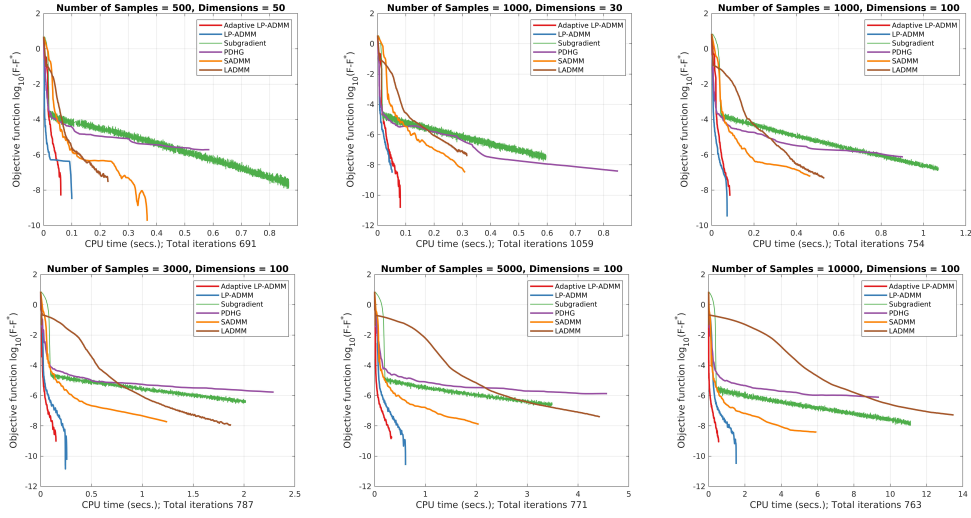

select the parameters of the DRLR model (i.e., $\epsilon = 0.3, \kappa = 7$). In addition, we randomly select 60% of the data to train the models and the rest to test the performance. As before, all the results reported here were averaged over 30 independent trials. Table 3 shows the average classification accuracy on the test data. We observe that the DRLR model consistently outperforms the two classic models over all datasets. Thus, the distributionally robust optimization approach opens a new door for ameliorating the poor test performance in practice.

Table 3: Average classification accuracy on test datasets

| Dataset | LR | RLR ($\kappa = \infty$) | DRLR ($\kappa = 7$) |
|---|---|---|---|
| a1a | 83.13% | 83.82% | **84.01%** |
| a2a | 83.68% | 83.93% | **84.24%** |
| MNIST(0 vs 3) | 99.15% | 99.45% | **99.55%** |
| MNIST(0 vs 4) | 99.39% | 99.54% | **99.75%** |
| MNIST(0 vs 6) | 97.88% | 98.86% | **98.92%** |
| MNIST(2 vs 3) | 96.87% | 97.31% | **97.40%** |
| MNIST(2 vs 5) | 96.67% | 97.45% | **97.77%** |
| MNIST(5 vs 8) | 94.80% | 94.91% | **95.18%** |
| MNIST(5 vs 9) | 97.21% | 98.11% | **98.47%** |
| MNIST(6 vs 9) | 99.54% | 99.59% | **99.80%** |

# 6 Conclusion and Future Work

In this paper, we have proposed a first-order algorithmic framework to solve a class of Wasserstein distance-based distributionally robust logistic regression (DRLR) problem. The core step of our framework is the efficient solution of the $\beta$-subproblem. Towards that end, we have developed a novel ADMM-type algorithm (the LP-ADMM) and established its sublinear rate of convergence. We have also conducted extensive experiments to verify the practicality of our framework. It is worth noting that problem (1.2) actually enjoys the Luo-Tseng error bound property when the transport cost is induced either by the $\ell_1$-norm or $\ell_\infty$-norm [16]. However, this does not immediately imply the linear convergence of our proposed LP-ADMM, as the method involves both primal and dual updates. Thus, it is interesting to see whether our proposed LP-ADMM or some other practically efficient first-order methods can provably achieve a linear rate of convergence when applied to the $\beta$-subproblem.

## Footnotes

[1] `https://www.csie.ntu.edu.tw/~cjlin/libsvmtools/datasets/binary.html`

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
