[Supplementary Material]

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

# Appendix

This supplementary document is the appendix section of the paper titled "First-Order Algorithmic Framework for Wasserstein Distributionally Robust Logistic Regression". It is organized as follows. In Section A, we present several technical lemmas and complete the proof of the main convergence result for the LP-ADMM. In Section B, we give the implementation details of our algorithmic framework, including pseudo-codes of the inner solvers and baseline algorithms. In Section C, we provide additional experiment results to support our findings in this paper.

## A : Technical Proof

Note that the objective function in problem (A) takes the form

$$\Omega(\lambda, \beta) = \lambda\epsilon + \frac{1}{N}\sum_{i=1}^{N}\left(h(\hat{y}_i\beta^T\hat{x}_i) + \max\{\hat{y}_i\beta^T\hat{x}_i - \lambda\kappa, 0\}\right) + \mathbb{I}_{\{\|\beta\|_\infty \leq \lambda\}}.$$

As the function $\Omega(\cdot, \cdot)$ is jointly convex, it is natural to conclude that the function $\lambda \mapsto q(\lambda) = \inf_\beta \Omega(\lambda, \beta)$ is also convex (and hence unimodal) on $\mathbb{R}$. The following lemma shows that this is indeed the case.

**Lemma 6.1.** *Suppose that $\Omega : \mathcal{X} \times \mathcal{Y} \longrightarrow \mathbb{R} \cup \{+\infty\}$ is a convex function that is proper and bounded below. Then, the function $x \mapsto q(x) = \inf_{y \in \mathcal{Y}} \Omega(x, y)$ is also convex.*

Recall that the $\beta$-subproblem of Wasserstein DRLR with $\ell_p$-induced transport cost (where $p \in \{1, 2, \infty\}$) is

$$
\begin{aligned}
\min_{\beta,\, s,\, \lambda} \quad & \lambda\epsilon + \frac{1}{N}\sum_{i=1}^{N} s_i \\
\text{s.t.} \quad & \ell_\beta(\hat{x}_i, \hat{y}_i) \leq s_i, \; i \in [N], \\
& \ell_\beta(\hat{x}_i, -\hat{y}_i) - \lambda\kappa \leq s_i, \; i \in [N], \\
& \|\beta\|_q \leq \lambda,
\end{aligned}
\tag{6.1}
$$

where $q$ satisfies $\frac{1}{p} + \frac{1}{q} = 1$.

**Proposition 6.2.** *Suppose that $(\beta^*, \lambda^*, s^*)$ is an optimal solution to (6.1). Then, we have $\lambda^* \leq \lambda^U = \frac{0.2785}{\epsilon}$.*

*Proof.* We analyze the case where $p = 1$ and other two cases are similar. From the KKT system in the proof of Proposition 3.1, we deduce

$$
\begin{aligned}
0 &= \sum_{i=1}^{N} a_{i1}\nabla_\beta\ell_\beta(\hat{x}_i, \hat{y}_i) + a_{i2}\nabla_\beta\ell_\beta(\hat{x}_i, -\hat{y}_i) + (a_{i3} - a_{i4})e_i \\
&= \sum_{i=1}^{N}\frac{1}{N}\nabla_\beta\ell_\beta(\hat{x}_i, \hat{y}_i) + a_{i2}(\nabla_\beta\ell_\beta(\hat{x}_i, -\hat{y}_i) - \nabla_\beta\ell_\beta(\hat{x}_i, \hat{y}_i)) + \sum_{i=1}^{N}(a_{i3} - a_{i4})e_i \\
&= \sum_{i=1}^{N}(\frac{1}{N}\nabla_\beta\ell_\beta(\hat{x}_i, \hat{y}_i) + a_{i2}\hat{y}_i\hat{x}_i) + \sum_{i=1}^{N}(a_{i3} - a_{i4})e_i.
\end{aligned}
$$

Multiplying $\beta$ gives

$$
\begin{aligned}
0 &= \sum_{i=1}^{N}(\frac{1}{N}\beta^T\nabla_\beta\ell_\beta(\hat{x}_i,\hat{y}_i) + a_{i2}\hat{y}_i\beta^T\hat{x}_i) + \sum_{i=1}^{N}(a_{i3}-a_{i4})e_i^T\beta \\
&= \sum_{i=1}^{N}(\frac{1}{N}\beta^T\nabla_\beta\ell_\beta(\hat{x}_i,\hat{y}_i) + a_{i2}\hat{y}_i\beta^T\hat{x}_i) + \lambda\sum_{i=1}^{N}(a_{i3}+a_{i4}) \\
&= \sum_{i=1}^{N}(\frac{1}{N}\beta^T\nabla_\beta\ell_\beta(\hat{x}_i,\hat{y}_i) + a_{i2}\hat{y}_i\beta^T\hat{x}_i) + \lambda(\epsilon-\kappa\sum_{i=1}^{N}a_{i2}) \\
&= \sum_{i=1}^{N}(\frac{1}{N}\beta^T\nabla_\beta\ell_\beta(\hat{x}_i,\hat{y}_i) + a_{i2}(\hat{y}_i\beta^T\hat{x}_i - \lambda\kappa)) + \lambda\epsilon \\
&= \sum_{i=1}^{N}(\frac{1}{N}\beta^T\nabla_\beta\ell_\beta(\hat{x}_i,\hat{y}_i) + a_{i2}(s_i - \ell_\beta(\hat{x}_i,\hat{y}_i))) + \lambda\epsilon \\
&\geq \frac{1}{N}\sum_{i=1}^{N}\beta^T\nabla_\beta\ell_\beta(\hat{x}_i,\hat{y}_i) + \lambda\epsilon.
\end{aligned}
$$

Plugging in the explicit formula for $\ell_\beta(\cdot,\cdot)$ yields

$$
\lambda \leq \frac{1}{N\epsilon}\sum_{i=1}^{N}\frac{\hat{y}_i\beta^T\hat{x}_i\exp(-\hat{y}_i\beta^T\hat{x}_i)}{1+\exp(-\hat{y}_i\beta^T\hat{x}_i)}. \tag{6.2}
$$

Note that $\phi(t) = \frac{te^{-t}}{1+e^{-t}} = \frac{t}{e^t+1}$ and $\phi'(t) = \frac{e^t-te^t+1}{(e^t+1)^2}$ is strictly decreasing. Thus, $\phi(t)$ has a unique maximizer and $\phi(t) \leq 0.2785$. Therefore, $\lambda \leq \lambda^U = \frac{0.2785}{\epsilon}$. $\qquad\square$

**Convergence Analysis of LP-ADMM**

**Lemma 6.3.** *The sequence $\{(x^k,y^k,z^k)\}_{k\geq 0}$ generated by the LP-ADMM satisfies*

$$
\|Ax^{k+1} - y^k\|_2^2 \geq \frac{1}{2}\|y^{k+1}-y^k\|_2^2 - \frac{L_f^2}{\rho^2}\|y^k-y^{k-1}\|_2^2
$$

*for all $k \geq 1$.*

*Proof.* Note that the $y$-update rule (3) takes the form

$$
y^{k+1} = \arg\min_{y}\left\{\frac{\rho}{2}\|y - (Ax^{k+1} + \frac{w^k+\nabla f(y^k)}{\rho})\|_2^2 + P(y)\right\}, \tag{6.3}
$$

which can be further rewritten as

$$
y^{k+1} = \text{prox}_{\frac{1}{\rho}g(.)}(Ax^{k+1} - \frac{w^k+\nabla f(y^k)}{\rho}). \tag{6.4}
$$

Based on the expansiveness property of proximal operator and the $y$-update rule (6.4), we have

$$
\begin{aligned}
\|y^{k+1}-y^k\|_2^2 &= \|\text{prox}_{\frac{1}{\rho}g(.)}(Ax^{k+1} - \frac{w^k+\nabla f(y^k)}{\rho}) - \text{prox}_{\frac{1}{\rho}g(.)}(Ax^k - \frac{w^{k-1}+\nabla f(y^{k-1})}{\rho})\|_2^2 \\
&\leq \|Ax^{k+1} - Ax^k + \frac{w^{k-1}-w^k}{\rho} + \frac{\nabla f(y^k)-\nabla f(y^{k-1})}{\rho}\|_2^2 \\
&\overset{(\heartsuit)}{=} \|Ax^{k+1} - y^k + \frac{1}{\rho}(\nabla f(y^k)-\nabla f(y^{k-1}))\|_2^2 \\
&\leq 2(\|Ax^{k+1}-y^k\|_2^2 + \frac{L_f^2}{\rho^2}\|y^k-y^{k-1}\|_2^2),
\end{aligned}
$$

where $(\heartsuit)$ is derived from $w^{k+1} = w^k - \rho(Ax^{k+1}-y^{k+1})$. Rearranging the inequality, we can conclude the proof. $\qquad\square$

**Lemma 6.4.** *Let $\{(x^k, y^k, w^k)\}_{k \geq 0}$ be the sequence generated by the LP-ADMM. Set $c = \frac{\rho - 2L_f}{4}$. Then, for any $x, y$ satisfying $Ax - y = 0$, we have*

$$F(x^{k+1}, y^{k+1}) - F(x, y) \leq \frac{1}{2\rho}(\|w^k\|_2^2 - \|w^{k+1}\|_2^2)$$

$$+ \frac{\rho}{2}(\|y^k - y\|_2^2 - \|y^{k+1} - y\|_2^2) + c(\|y^k - y^{k-1}\|_2^2 - \|y^{k+1} - y^k\|_2^2) + (B_f(y, y^{k+1}) - B_f(y, y^k)).$$

*Proof.* Using the optimality conditions (4.1) and (4.2), we have

$$- \rho A^T(Ax^{k+1} - y^k - \frac{w^k}{\rho}) \in \partial g(x^{k+1}), \tag{6.5}$$

$$\nabla f(y^{k+1}) - \nabla f(y^k) - w^{k+1} \in \nabla f(y^{k+1}) + \partial P(y^{k+1}), \tag{6.6}$$

where (6.6) is derived from $w^{k+1} = w^k - \rho(Ax^{k+1} - y^{k+1})$. By the convexity of $g(\cdot)$,

$$g(x^{k+1}) - g(x) \leq \langle \partial g(x^{k+1}), x^{k+1} - x \rangle$$

$$= -\langle \rho A^T(Ax^{k+1} - y^k - \frac{w^k}{\rho}), x^{k+1} - x \rangle$$

$$= -\rho \langle Ax^{k+1} - y^{k+1} + y^{k+1} - y^k - \frac{w^k}{\rho}, A(x^{k+1} - x) \rangle$$

$$= -\rho \langle \frac{1}{\rho}(w^k - w^{k+1}) + y^{k+1} - y^k - \frac{w^k}{\rho}, A(x^{k+1} - x) \rangle$$

$$= \langle w^{k+1} - \rho(y^{k+1} - y^k), Ax^{k+1} - y \rangle \quad \text{(i.e., Dual Update and Dual Feasibility)}$$

$$= \langle w^{k+1}, Ax^{k+1} - y \rangle + \langle \rho(y^k - y^{k+1}), Ax^{k+1} - y \rangle$$

$$= \langle w^{k+1}, Ax^{k+1} - y \rangle + \frac{\rho}{2}(\|y^k - y\|_2^2 - \|y^{k+1} - y\|_2^2 + \|Ax^{k+1} - y^{k+1}\|_2^2$$

$$- \|Ax^{k+1} - y^k\|_2^2),$$

where the last equality holds as

$$(x_1 - x_2)^T(x_3 + x_4) = \frac{1}{2}(\|x_4 - x_2\|_2^2 - \|x_4 - x_1\|_2^2 + \|x_3 + x_1\|_2^2 - \|x_3 - x_2\|_2^2).$$

Similarly, by the convexity of the function $f(\cdot) + P(\cdot)$, we have

$$f(y^{k+1}) + P(y^{k+1}) - f(y) - P(y) \leq \langle \nabla f(y^{k+1}) + \partial P(y^{k+1}), y^{k+1} - y \rangle$$

$$= \langle \nabla f(y^{k+1}) - \nabla f(y^k) - w^{k+1}, y^{k+1} - y \rangle$$

$$= \langle \nabla f(y^{k+1}) - \nabla f(y^k), y^{k+1} - y \rangle - \langle w^{k+1}, y^{k+1} - y \rangle.$$

Summing the above two inequalities, we have

$$F(x^{k+1}, y^{k+1}) - F(x, y) \leq \langle w^{k+1}, Ax^{k+1} - y^{k+1} \rangle + \frac{\rho}{2}\|Ax^{k+1} - y^{k+1}\|_2^2$$

$$+ \frac{\rho}{2}(\|y^k - y\|_2^2 - \|y^{k+1} - y\|_2^2)$$

$$+ \underbrace{\langle \nabla f(y^{k+1}) - \nabla f(y^k), y^{k+1} - y \rangle - \frac{\rho}{2}\|Ax^{k+1} - y^k\|_2^2}_{(*)}.$$

Note that the term $\langle w^{k+1}, Ax^{k+1} - y^{k+1} \rangle + \frac{\rho}{2}\|Ax^{k+1} - y^{k+1}\|_2^2$ can be reformulated as $\frac{1}{2\rho}(\|w^k\|_2^2 - \|w^{k+1}\|_2^2)$ via plugging in the dual update rule. In details,

$$\langle w^{k+1}, Ax^{k+1} - y^{k+1} \rangle + \frac{\rho}{2}\|Ax^{k+1} - y^{k+1}\|_2^2 = \langle Ax^{k+1} - y^{k+1}, w^{k+1} + \frac{\rho}{2}(Ax^{k+1} - y^{k+1}) \rangle$$

$$= \langle \frac{w^k - w^{k+1}}{\rho}, \frac{1}{2}(w^k + w^{k+1}) \rangle$$

$$= \frac{1}{2\rho}(\|w^k\|_2^2 - \|w^{k+1}\|_2^2).$$

Besides, it is more tricky to give an upper bound on the last term in the above gap function. Based on the three-point property of Bregman divergence, we have

$$\langle \nabla f(y^{k+1}) - \nabla f(y^k), y^{k+1} - y \rangle = B_f(y^{k+1}, y^k) + B_f(y, y^{k+1}) - B_f(y, y^k)$$
$$\leq \frac{L_f}{2}\|y^{k+1} - y^k\|_2^2 + (B_f(y, y^{k+1}) - B_f(y, y^k)). \tag{6.7}$$

Applying Lemma 6.3 to the term $-\frac{\rho}{2}\|Ax^{k+1} - y^k\|_2^2$, we have

$$-\frac{\rho}{2}\|Ax^{k+1} - y^k\|_2^2 \leq \frac{L_f^2}{2\rho}\|y^k - y^{k-1}\|_2^2 - \frac{\rho}{4}\|y^{k+1} - y^k\|_2^2. \tag{6.8}$$

Adding (6.7) and (6.8), we can obtain

$$(*) \leq \frac{L_f^2}{2\rho}\|y^k - y^{k-1}\|_2^2 - (\frac{\rho}{4} - \frac{L_f}{2})\|y^{k+1} - y^k\|_2^2 + (B_f(y, y^{k+1}) - B_f(y, y^k)). \tag{6.9}$$

If the penalty parameter satisfies the condition $\rho \geq (\sqrt{3} + 1)L_f$, then

$$(*) \leq c(\|y^k - y^{k-1}\|_2^2 - \|y^{k+1} - y^k\|_2^2) + (B_f(y, y^{k+1}) - B_f(y, y^k)). \tag{6.10}$$

$\square$

**Proposition 6.5.** *Suppose that we use a constant penalty parameter $\rho$ that satisfies $\rho > (\sqrt{3} + 1)L_f$. Let $\{(x^k, y^k, w^k)\}_{k \geq 0}$ be the sequence generated by the LP-ADMM and $(x^*, y^*, w^*)$ be a point satisfying the KKT conditions (2.3) with $x^* \in \mathcal{X}, y^* \in \mathcal{Y}$. Then, the following hold:*

(a) *For all $k \geq 1$, $\|Ax^{k+1} - y^k\|_2^2 \geq \frac{1}{2}\|y^{k+1} - y^k\|_2^2 - \frac{L_f^2}{\rho^2}\|y^k - y^{k-1}\|_2^2$.*

(b) *For all $k \geq 0$ and $(x, y)$ satisfying $Ax - y = 0$, we have $F(x^{k+1}, y^{k+1}) - F(x, y) \leq \frac{1}{2\rho}(\|w^k\|_2^2 - \|w^{k+1}\|_2^2) + \frac{\rho}{2}(\|y^k - y\|_2^2 - \|y^{k+1} - y\|_2^2) + c(\|y^k - y^{k-1}\|_2^2 - \|y^{k+1} - y^k\|_2^2) + (B_f(y, y^{k+1}) - B_f(y, y^k))$, where $c = \frac{\rho - 2L_f}{4}$.*

(c) *The sequence $\{\frac{1}{2\rho}\|w^k - w^*\|_2^2 + \frac{\rho}{2}\|y^k - y^*\|_2^2 - B_f(y^*, y^k)\}_{k \geq 0}$ is non-increasing and bounded below.*

*Proof.* Parts (a) and (b) have been already proven in Lemmas 6.2 and 6.3. By the convexity of $F(\cdot, \cdot)$,

$$F(x^*, y^*) - F(x^{k+1}, y^{k+1}) \leq -\langle w^*, Ax^{k+1} - y^{k+1} \rangle. \tag{6.11}$$

Subsequently, combining (6) and (6.7),

$$F(x^{k+1}, y^{k+1}) - F(x^*, y^*) \leq \langle w^{k+1}, Ax^{k+1} - y^{k+1} \rangle + \frac{\rho}{2}\|Ax^{k+1} - y^{k+1}\|_2^2$$
$$+ \frac{\rho}{2}(\|y^k - y^*\|_2^2 - \|y^{k+1} - y^*\|_2^2) - (B_f(y^*, y^k) - B_f(y^*, y^{k+1}))$$
$$+ \frac{L_f}{2}\|y^{k+1} - y^k\|_2^2 - \frac{\rho}{2}\|Ax^{k+1} - y^k\|_2^2. \tag{6.12}$$

Summing up the terms (6.11) and (6.12), we have

$$0 \leq \frac{1}{\rho}\langle w^{k+1} - w^*, w^k - w^{k+1} \rangle + \frac{\rho}{2}\|Ax^{k+1} - y^{k+1}\|_2^2 + \frac{\rho}{2}(\|y^k - y^*\|_2^2 - \|y^{k+1} - y^*\|_2^2)$$
$$- (B_f(y^*, y^k) - B_f(y^*, y^{k+1})) + \frac{L_f}{2}\|y^{k+1} - y^k\|_2^2 - \frac{\rho}{2}\|Ax^{k+1} - y^k\|_2^2. \tag{6.13}$$

Note that $m_{k+1} = \frac{1}{2\rho}\|w^{k+1} - w^*\|_2^2 + \frac{\rho}{2}\|y^{k+1} - y^*\|_2^2 - B_f(y^*, y^{k+1})$. Based on $(x_1 - x_2)^T(x_1 - x_3) = \frac{1}{2}(\|x_1 - x_2\|_2^2 + \|x_1 - x_3\|_2^2 - \|x_2 - x_3\|_2^2)$, we have,

$$0 \leq m_k - m_{k+1} + \frac{L_f}{2}\|y^{k+1} - y^k\|_2^2 - \frac{\rho}{2}\|Ax^{k+1} - y^k\|_2^2. \tag{6.14}$$

On top of Lemma 6.3,

$$(-\frac{L_f^2}{2\rho} + \frac{\rho}{4} - \frac{L_f}{2})\|y^{k+1} - y^k\|_2^2 \le m_k - m_{k+1} + \frac{L_f^2}{2\rho}(\|y^k - y^{k-1}\|_2^2 - \|y^{k+1} - y^k\|_2^2). \quad (6.15)$$

We construct a new sequence $m'_{k+1} = m_{k+1} + \frac{L_f^2}{2\rho}\|y^{k+1} - y^k\|_2^2$. Since the term $-\frac{L_f^2}{2\rho} + \frac{\rho}{4} - \frac{L_f}{2}$ is positive if $\rho > (\sqrt{3} + 1)L_f$, it is easy to conclude that the sequence $m'_{k+1}$ is non-increasing and bounded below. Moreover, we have

$$\sum_{k=0}^{\infty} \|y^{k+1} - y^k\|_2^2 < +\infty.$$

Thus, we conclude that the sequence $\{m_k\}$ is non-increasing, bounded below and $\|y^{k+1} - y^k\| \to 0$. Due to the dual update rule, we automatically have $\|x^{k+1} - x^k\| \to 0$ and $\|w^{k+1} - w^k\| \to 0$. $\square$

**Theorem 6.6.** *Consider the setting of Proposition 6.5. Set $\bar{x}^K = \frac{1}{K}\sum_{k=1}^{K} x^k$ and $\bar{y}^K = \frac{1}{K}\sum_{k=1}^{K} y^k$. Then, the following hold:*

*(a) The sequence $\{(x^k, y^k, w^k)\}_{k\ge 0}$ converges to a KKT point of problem (2.1).*

*(b) The sequence of function values converges at the rate $\mathcal{O}(\frac{1}{K})$:*

$$F(\bar{x}^K, \bar{y}^K) - F(x^*, y^*) \le \frac{(1/2\rho)\|w^0\|_2^2 + (\rho/2)\|y^* - y^0\|_2^2 + c\|y^0 - y^1\|_2^2}{K} = \mathcal{O}(\frac{1}{K}).$$

*Proof.* (a): By Proposition 6.5, we know that $\{m_k\}$ is a convergent sequence. Thus, $\{(x^k, y^k, w^k)\}$ is a bounded sequence. Every bounded sequence in $\mathbb{R}^n$ contains a convergent subsequence. Note that $\{(x^{k_j}, y^{k_j}, w^{k_j})\}$ is the corresponding subsequence. Hence, it has a limit point, which we denote by $(x^\infty, y^\infty, w^\infty)$.

- **Step 1:** Prove that the accumulation point $(x^\infty, y^\infty, w^\infty)$ is a KKT point.

As the dual update $\frac{1}{\rho}(w^k - w^{k+1}) = Ax^{k+1} - y^{k+1}$ and $\|w^k - w^{k+1}\| \to 0$, we have $\|Ax^{k+1} - y^{k+1}\| \to 0$ and $Ax^\infty - y^\infty = 0$. Thus, any accumulation point is dual feasible. Moreover, by the convexity of $F(\cdot, \cdot)$ and the optimality conditions (4.1) and (4.2),

$$F(x^*, y^*) \ge F(x^{k_j}, y^{k_j}) - \langle \rho A^T(Ax^{k_j+1} - y^{k_j} - \frac{w^{k_j}}{\rho}), x^* - x^{k_j}\rangle \\ + \langle \nabla f(y^{k_j+1}) - \nabla f(y^{k_j}) - w^{k_j+1}, y^* - y^{k_j}\rangle. \quad (6.16)$$

As $j \to \infty$, we have

$$F(x^*, y^*) \ge F(x^\infty, y^\infty) - \langle w^\infty, A(x^* - x^\infty) + y^\infty - y^*\rangle. \quad (6.17)$$

Since the point $(x^\infty, y^\infty)$ is dual feasible, $F(x^*, y^*) \ge F(x^\infty, y^\infty)$. As $(x^\infty, y^\infty)$ and $(x^*, y^*)$ are both feasible solutions, $(x^\infty, y^\infty)$ is an optimal solution.

Moreover, based on the convexity of $g(\cdot)$ and the $x$-update optimality condition (4.1), we have

$$g(x) \ge g(x^{k_j}) - \langle \rho A^T(Ax^{k_j+1} - y^{k_j} - \frac{w^{k_j}}{\rho}), x - x^{k_j}\rangle. \quad (6.18)$$

As $j \to \infty$, we have

$$g(x) \ge g(x^\infty) + \langle A^T w^\infty, x - x^\infty\rangle. \quad (6.19)$$

Thus, $A^T w^\infty \in \partial g(x^\infty)$ and similarly $-w^\infty \in \nabla f(y^\infty) + \partial P(y^\infty)$. Therefore, $(x^\infty, y^\infty, w^\infty)$ is a KKT point.

- **Step 2:** Prove that the whole sequence $\{(x^k, y^k, w^k)\}_{k\ge 0}$ converges to $(x^\infty, y^\infty, w^\infty)$.

By choosing $(x^*, y^*, w^*) = (x^\infty, y^\infty, w^\infty)$ in Proposition 6.5, we have $\frac{1}{2\rho}\|w^{k_j+1} - w^\infty\|_2^2 + \frac{\rho}{2}\|y^{k_j+1} - y^\infty\|_2^2 - B_f(y^\infty, y^{k_j+1}) \to 0$. Subsequently, by Proposition 6.5(a), it is easy to conclude that $\frac{1}{2\rho}\|w^{k+1} - w^\infty\|_2^2 + \frac{\rho}{2}\|y^{k+1} - y^*\|_2^2 - B_f(y^\infty, y^{k+1}) \to 0$. Therefore, $(x^k, y^k, w^k) \to (x^\infty, y^\infty, w^\infty)$. As the point $(x^\infty, y^\infty, w^\infty)$ can be any limit point of the sequence $\{(x^k, y^k, w^k)\}_{k \geq 0}$, we conclude that $(x^\infty, y^\infty, w^\infty)$ is a KKT point.

(b): On top of Proposition 6.5(b), we have

$$F(x^{k+1}, y^{k+1}) - F(x^*, y^*) \leq \frac{1}{2\rho}(\|w^k\|_2^2 - \|w^{k+1}\|_2^2) + \frac{\rho}{2}(\|y^k - y^*\|_2^2 - \|y^{k+1} - y^*\|_2^2)$$
$$+ c(\|y^k - y^{k-1}\|_2^2 - \|y^{k+1} - y^k\|_2^2) + (B_f(y^*, y^{k+1}) - B_f(y^*, y^k)).$$

Since $F(\cdot, \cdot)$ is convex, by the Jensen inequality, we have

$$F(\bar{x}^K, \bar{y}^K) - F(x^*, y^*) \leq \frac{1}{K}\sum_{k=1}^{K}(F(x^k, y^k) - F(x^*, y^*))$$

$$\leq \frac{1}{K}\left\{\frac{1}{2\rho}(\|w^0\|_2^2 - \|w^K\|_2^2) + \frac{\rho}{2}(\|y^0 - y^*\|_2^2 - \|y^K - y^*\|_2^2) + c(\|y^1 - y^0\|_2^2 - \|y^K - y^{K-1}\|_2^2)\right\}$$

$$\leq \frac{1}{K}\left\{\frac{1}{2\rho}\|w^0\|_2^2 + \frac{\rho}{2}\|y^0 - y^*\|_2^2 + c\|y^1 - y^0\|_2^2\right\}.$$

Noting that $B_f(y^*, y^K) - B_f(y^*, y^0) < 0$ when $K$ is sufficiently large, we complete the proof. $\square$

**B: Implementation Details**

Figure 4: Schematics of the Golden Section Search Method

---

**Algorithm 2:** Golden Section Search Algorithm

---

**Input**: $\lambda_1 = 0$, $\lambda_4 = \lambda^U$, $r = 0.618$;
**Output**: Optimal solution $(\beta^*, \lambda^*)$;
1 **while** *not converge* **do**
2    $\lambda_2 = r\lambda_1 + (1 - r)\lambda_4$;
3    $\lambda_3 = (1 - r)\lambda_1 + r\lambda_4$;
4    $q(\lambda_1) = \text{LP-ADMM}(\lambda_1)$;
5    $q(\lambda_2) = \text{LP-ADMM}(\lambda_2)$;
6    $q(\lambda_3) = \text{LP-ADMM}(\lambda_3)$;
7    $q(\lambda_4) = \text{LP-ADMM}(\lambda_4)$;
8    **if** $q(\lambda_2) < q(\lambda_3)$ **then**
9       set $\lambda_4 = \lambda_3$
10   **else**
11       set $\lambda_1 = \lambda_2$
12   **end**
13 **end**

---

**Primal Dual Hybrid Gradient Method**

$$\min_{\|x\|_\infty \leq \lambda} \max_{\|y\|_\infty \leq 1} \frac{1}{N} \sum_{i=1}^{N} \left\{ \log(1 + \exp(-a_i^T x)) + \frac{1}{2}(a_i^T x - b_i) \right\} + \frac{1}{2N} y^T (Ax - b) \quad (6.20)$$

**Standard ADMM**

$$
\begin{aligned}
\min_{x,\,y} \quad & f(y) + g(z) + \mathbb{I}_{\{\|x\|_\infty \leq \lambda\}} \\
\text{s.t.} \quad & Ax = y, \\
& z = y - b.
\end{aligned}
\quad (6.21)
$$

The corresponding augmented Lagrangian function is

$$\mathcal{L}_\rho(x, y; u, v) = f(y) + g(z) + u^T(Ax - y) + v^T(z - y + b) + \frac{\rho}{2}\|Ax - y\|_2^2 + \frac{\rho}{2}\|z - y + b\|_2^2.$$

**Remark 6.7.** *We can regard $(y, z)$ as a block in problem* (6.21) *and apply the standard ADMM to solve it. Moreover, though $g(\cdot)$ is a non-smooth function, it statisfies the semi-smooth property and its generalized Hessian matrix can be easily computed due to the log-loss function. Namely,*

$$(y^{k+1}, z^{k+1}) = \arg\min_y \left\{ f(y) + \frac{1}{2}\|y - d_1\|^2 + \min_z\{g(z) + \frac{1}{2}\|z - (y - d_2)\|^2\} \right\}, \quad (6.22)$$

*where $d_1 = Ax^{k+1} + \frac{\mu^k}{\rho}$ and $d_2 = b + \frac{v^k}{\rho}$.*

---

**Algorithm 3:** Standard ADMM

---

**Input**: $\rho = 10$, $\sigma = 1$, $k = 1$, randomly generate $x^1, y^1, z^1, u^1, v^1$;
**Output**: Optimal solution $(x^*, y^*, z^*)$;

1 **while** *not converge* **do**

2 $\quad x^{k+1} = \underset{x}{\arg\min} \left\{ \frac{\rho}{2}\|Ax - y^k + \frac{u^k}{\rho}\|_2^2 + g(x) \right\}$ ;  $\quad\quad\quad\quad \triangleright$ `Conjugate Gradient Method`

3 $\quad y^{k+1} = \underset{y \in \mathbb{R}^N}{\arg\min} \left\{ f(y) + \frac{\rho}{2}\|y - Ax^{k+1} - \frac{u^k}{\rho}\|_2^2 \right\}$;  $\quad\quad\quad\quad \triangleright$ `Semi-Smooth Newton`

4 $\quad z^{k+1} = \underset{z \in \mathbb{R}^N}{\arg\min} \left\{ g(z) + \frac{\rho}{2}\|z - y^{k+1} + b + \frac{v^k}{\rho}\|_2^2 \right\}$;

5 $\quad u^{k+1} = u^k + \sigma\rho(Ax^{k+1} - y^{k+1})$;

6 $\quad v^{k+1} = v^k + \sigma\rho(z^{k+1} - y^{k+1} + b)$;

7 $\quad k = k + 1$;

8 **end**

---

**Algorithm 4:** Semi-smooth Newton Method

---

**Input**: $k = 1$, random generate $y^1$;
**Output**: Optimal solution $y^*$;

1 **while** *not converge* **do**

2 $\quad \nabla f(y^k) = -\frac{e^{-y^k}}{(1+e^{-y^k})} + \rho(y - Ax - \frac{u}{\rho}) + \rho(y - z - b - \frac{v}{\rho})$;

3 $\quad$ **if** $y_i^k > (b + \frac{v}{\rho})_i$ *or* $y_i^k < (b + \frac{v}{\rho})_i + \frac{1}{\rho}$ **then**

4 $\quad\quad h_i = 1$;

5 $\quad$ **else**

6 $\quad\quad h_i = 0$;

7 $\quad$ **end**

8 $\quad \nabla^2 f(y^k) = \frac{e^{-y^k}}{(1+e^{-y^k})^2} + \rho + \rho h$;  $\quad\quad\quad\quad \triangleright$ `computing the generalized Hessian`

9 $\quad g^k = (\nabla^2 f(y^k))^{-1} \cdot \nabla f(y^k)$;

10 $\quad y^{k+1} = y^k + ss \cdot g^k$;  $\quad\quad\quad\quad \triangleright$ `ss is returned by line search`

11 $\quad k = k + 1$;

12 **end**

---

Note that the subproblem solvers are the accelerated projected gradient method and semi-smooth Newton method, respectively.

**Solver for Quadratic Minimization with Box constraints**

$$\min_x \quad \|Ax - b\|_2^2$$
$$\text{s.t.} \quad \|x\|_\infty \le \lambda. \tag{6.23}$$

---

**Algorithm 5:** Accelerated Projected Gradient

**Input**: $ss = 1/\lambda_{max}(A)$, $k = 1$, random generate $x^1 = x_{old}$;
**Output**: Optimal solution $x^*$;
1 **while** *not converge* **do**
2     $\beta_k = \frac{k}{k+3}$;
3     $y^k = x^k + \beta(x^k - x_{old})$;
4     $g^k = A^T(Ax^k - b)$;
5     $w^k = y^k - ss \cdot g^k$;
6     $w^k = \text{proj}_{\{\|w\|_\infty \le \lambda\}}(w^k)$;
7     $x_o = x^k$;
8     $x^k = w^k$;
9     $k = k + 1$;
10 **end**

---

**Algorithm 6:** Conjugate Gradient with Active Set

**Input**: Randomly initialize $x^1$, $g^1 = A^T(Ax^1 - b)$, $k = 1$;
Bound set $\bar{B} = \{i : |x_i| = \lambda \text{ and } -g_i \cdot x_i \ge 0\}$, Free set $\bar{F} = \{i : i \notin \bar{B}\}$;
$r_i^k = \begin{cases} -g_i^k, & i \in \bar{F}, \\ 0, & \text{otherwise} \end{cases}$
**Output**: Optimal solution $x^*$;
1 **while** *not converge* **do**
2     Update $\bar{F}$;
3     $r_i^k = \begin{cases} -g_i^k, & i \in \bar{F}, \\ 0, & \text{otherwise} \end{cases}$;
4     **if** $k = 1$ or $\bar{F}$ changed **then**
5        $p^k = r^k$
6     **else**
7        $\beta_k = \frac{\|r^k\|^2}{\|r^{k-1}\|^2}$;
8        $p^k = r^k + \beta_k p^{k-1}$
9     **end**
10     $\alpha_k = \frac{\|r^k\|^2}{\|Ap^k\|^2}$;
11     $\tilde{x}^k = x^{k-1} + \alpha_k p^k$;
12     $x^k = \text{proj}_{\{\|x\|_\infty \le \lambda\}}(\tilde{x}^k)$;
13     **if** $\tilde{x}^k = x^k$ **then**
14        $g^k = g^k + \alpha_k Ap^k$
15     **else**
16        $g^k = A^T(Ax^k - b)$
17     **end**
18     $k = k + 1$;
19 **end**

---

**Remark 6.8.** *As problem* (6.23) *takes a similar form as the dual formulation of SVM, (i.e., see the form (4) in [10]), we use the same coordinate minimization algorithm as our alternative inner solver.*

Table 4: The CPU time $(s)$ of our proposed first-order algorithmic framework on real datasets

| Dataset | Samples | Features | CPU time (s) | Sparsity Level |
|---|---|---|---|---|
| mushrooms | 8123 | 112 | 8.47 | 0.8125 |
| phishing | 11055 | 68 | 4.01 | 0.7543 |
| w1a | 2477 | 300 | 13.15 | 0.9618 |
| w2a | 3470 | 300 | 15.13 | 0.9612 |
| w3a | 4912 | 300 | 15.96 | 0.9612 |
| w4a | 7366 | 300 | 19.35 | 0.9611 |
| w5a | 9888 | 300 | 19.50 | 0.9612 |
| w6a | 17188 | 300 | 11.42 | 0.9611 |
| w7a | 24692 | 300 | 12.07 | 0.9611 |
| w8a | 49749 | 300 | 14.86 | 0.9612 |
| MNIST(0 vs 3) | 12054 | 752 | 118.68 | 0.7641 |
| MNIST(0 vs 4) | 11765 | 754 | 117.81 | 0.7784 |
| MNIST(0 vs 6) | 11841 | 720 | 118.79 | 0.7577 |
| MNIST(2 vs 3) | 12089 | 752 | 127.43 | 0.7792 |
| MNIST(2 vs 5) | 11379 | 771 | 134.80 | 0.7912 |
| MNIST(5 vs 8) | 11272 | 771 | 111.54 | 0.7883 |
| MNIST(5 vs 9) | 11370 | 780 | 123.89 | 0.8109 |
| MNIST(6 vs 9) | 11867 | 780 | 139.14 | 0.8077 |

## C: Additional Experiment Results

Here, the sparsity level is the ratio of the zero elements to the total elements. The stopping criterion used in our proposed LP-ADMM is the dual infeasibility $\|Ax^{k+1} - y^{k+1}\|_2 \leq 10^{-6}$ in all experiments.