[Reviews · NeurIPS 2019]

Reviewer 1



This paper derives a novel algorithm for solving the dual DRLR problem when \kappa < \infty (i.e. the labels may change during transport). The algorithm performs a golden section search for \lambda, within which the sub-problem for optimal \beta, fixing \lambda, is solved by an ADMM algorithm. The ADMM algorithm differs from typical ADMM approaches in two ways: (1) the \beta-update is ill-conditioned, requiring a careful choice of iterative method, while (2) the auxiliary \mu update is locally strongly convex, enabling the use of a first-order (not quadratic) approximation with a fixed step size. I see three theoretical contributions: 1. An upper bound on optimal \lambda, stated in Proposition 1, which enables the golden section search. 2. The LP-ADMM iterative method for solving the \beta subproblem, stated as Algorithm 1. 3. O(1/k) convergence of LP-ADMM, stated in Theorem 4.2. And two empirical results: 1. Substantial speedups (wall clock) over standard YALMIP solver on the full DRLR problem, for both adaptive and nonadaptive step size selection strategies for the inner \beta subproblem, using both synthetic and real data. 2. Fast convergence of LP-ADMM on the \beta subproblem, as compared to several standard iterative methods, using synthetic data. Strengths: 1. Fast solution of DRLR is potentially useful and (to my knowledge) previously unaddressed. 2. The paper is fairly easy to follow, with the exception of the notation change between sections. 3. I like Figure 1, which shows the relationship of LP-ADMM to the DRLR problem. 4. The empirical speedups appear to be substantial. Weaknesses: 1. The change in notation from the DRLR sections (1, 3, and 5) to the generic LP-ADMM sections (2 and 4) is potentially confusing. Is there a way to make the notation consistent? (Or at least disjoint?) 2. Section 5.3 bears little relation to the rest of the paper and seems to show a result nearly identical to Table 1 in [1] (using different datasets). Should it be omitted entirely? 3. Section 5.2 uses only a single synthetic problem to argue for faster convergence of LP-ADMM as compared to standard methods. Could you show something like number of iterations to convergence under various tolerances, using multiple real datasets? Or is there another way to show that the standard methods are problematic, given real data? This claim needs more evidence. 4. There's no analysis of the relative contributions of the golden section search versus the LP-ADMM. I.e. if I plug in a standard solver for the \beta subproblem (in place of LP-ADMM), does golden section search still yield a substantial speedup? And is there a reason to believe golden section search will outperform other univariate search methods? Less major weaknesses: 1. There's no real discussion of the choice between solvers for the \beta update within the ADMM. Which should I choose and when? Could you expand the discussion on lines 129-132? At least: What are the tradeoffs between the methods? (Presumably efficiency, under different assumptions.) Could you illustrate the difference empirically? Are there practical rules involving, e.g., condition numbers for the data matrices? 2. In Section 5.1, there's no mention of the chosen iterative method for the \beta update in the ADMM. 3. In Section 5.1, there's no comparison of adaptive to nonadaptive penalty using real data (Figure 3). Also, it's unclear which of these you're using in Figure 3. Typos, etc.: * The horizontal axis in Figure 2 isn't labeled. * There's a typo in Remark 6.9 (in the appendix): "... it statisfies ..." [1] Shafieezadeh-Abadeh, Esfahani, Kuhn. Distributionally Robust Logistic Regression. NIPS 2015. ---- UPDATE: I have read the authors' response and I appreciate their explanations and willingness to include additional content. I will not be changing my score.

Reviewer 2



Originality: The WDR-LR problem is timely and important as it can offer advantages over other logistic regression formulations, as demonstrated by previous works. The proposed framework for wrapping an ADMM method in a line search method to address this problem appears to an appropriate and novel approach. It is also a nice motivation for the development of the new LP-ADMM method variant. The main novelty of the LP-ADMM method is that the y variables uses a first-order instead of second-order approximation, which is justified by the strong convexity of that subproblem. While the change may appear somewhat minor, this leads to a more aggressive/robust strategy with demonstrated benefits based on the numerical experiments. The authors do a good job for the most part on the literature review, although they may want to discuss some more works on applying first order methods to different types of DRO problems than those studied herein, see [1] below for example. Clarity: The paper is for the most part written well and is well organized. Quality: To the best of my knowledge, the mathematical analysis and proofs are correct. Although the O(1/K) convergence rate does not improve on other types of related ADMM methods, the numerical experiments justify its superior performance for the WDR-LR problem. In particular, the authors demonstrate improved performance over both off he shelf YALMIP solver as well as other existing first order/ADMM methods for solving the beta subproblem. Significance: The WDR-LR problem is timely and important as it can offer advantages over other logistic regression formulation, as demonstrated by previous works. Still, since the problem is somewhat narrow and the formulation is more complicated than ridge logistic regression for example, it may be difficult for this work to have a very strong amount of practical impact and/or follow up works. Nevertheless, this paper appears to be an important first step in addressing Wasserstein DRO type problems with large-scale optimization algorithms. [1] Namkoong, Hongseok, and John C. Duchi. "Stochastic gradient methods for distributionally robust optimization with f-divergences." Advances in Neural Information Processing Systems. 2016.

Reviewer 3



1. It would benefit the reader if the result is stated for the general norm for beta, instead of merely mentioning "the framework is general enough to extend to other norms". 2. What is the relationship between the KKT point of the problem (2.1) and the global solution of the original problem? In particular, if we get an eps-optimal solution of (2.1), what is the optimality gap of the original DRLR problem? 3. DRO typically works better the empirical risk minimization when the sample size is relatively small. Several numerical experiments use a large dataset such as MNIST. Can you run the experiment with a much smaller subset of the data and make corresponding comparisons? 4. What is the information-theoretic lower bound of the DRLR problem? This is related to the last sentence in the conclusion section. 5. A minor comment: can you state the problem in terms of the original variables beta, lambda, and s, as it facilitates the reader?

[Author Response · NeurIPS 2019]

We thank the reviewers for their insightful comments. Below please find our responses to the major points raised.

**To Reviewers 1:**    We appreciate your very detailed and thoughtful comments.

**Q1**: Quantify the relative contributions of the golden section search and the LP-ADMM to the runtime speedup.

**A1**: The use of the golden section search alone will not lead to any substantial speedup, as the main computational
burden lies in the $\beta$-subproblem. In general, standard solvers (such as interior-point methods) cannot exploit the
structure of the $\beta$-subproblem. Thus, even when combined with the golden section search strategy, they still cannot
achieve the speedup obtained by our proposed LP-ADMM. Furthermore, the golden section search enjoys the $\Theta(\log \frac{1}{\epsilon})$
complexity, which is already optimal in the information-theoretic sense. Therefore, any other univariate search methods
can at best achieve similar complexity.

**Q2**: Strengthen Section 5.2 with further empirical evidence of faster convergence in a variety of settings.

**A2**: Thank you for your kind reminder. As mentioned in the paper (line 235-237), we did the comparison for the real
datasets (i.e., UCI Adult) but found that all baseline methods cannot achieve the desired accuracy (i.e.,$||x_k - x^*|| \leq$
$10^{-6}$). Due to the ill-conditioned data matrix, all baseline methods require an extremely careful choice of hyper-
parameters. That's why we did not include the details. Based on your advice, we will add back the comparison in the
revision.

**Q3**: The change in notation from the DRLR sections (1, 3, and 5) to the generic LP-ADMM sections (2 and 4) is
potentially confusing. Is there a way to make the notation consistent?

**A3**: Thank you for your suggestion. We will unify our notation for clarity in the revision.

**Q4**: Section 5.3 bears little relation to the rest of the paper ...... Should it be omitted entirely?

**A4**: Thank you for your comments. We want to further verify the power of DRO modeling for the large-scale datasets,
which is different from Table 1 in [1]. That is why we include it.

**Q5**: The choice of inner solvers for $\beta$-update in LP-ADMM for different settings (illustrate the difference empirically?)

**A5**: Thank you for your advice. We will add them in the main text in the revision for clarity.

**To Reviewers 2:**    Thanks for appreciating the contributions of our work. Thanks for the suggestion on the literature
review. We have discussed [1] in our revised manuscript.

**To Reviewers 3:**

**Q1**: Can you prove results and conduct experiments for different norm constraint on beta, such as $\ell_1$ or $\ell_2$? It would
benefit the reader if the result is stated for the general norm for beta.

**A1**: Thank you for your suggestion. Absolutely yes. Firstly, for an upper bound on optimal $\lambda$ for $\ell_1$ and $\ell_2$ cases, we
already have the same upper bound whose proof is just a slight modification of the current one in the appendix for the
$\ell_\infty$ case. In details, we replace the ball constraint $||\beta||_1 \leq \lambda$ by the equivalent one $B\beta \leq \lambda e_{2^n}$ where $B$ is the $2^n \times n$
matrix whose rows are all the possible arrangements of $+1's$ and $-1's$, and $||\beta||_2 \leq \lambda$ by $||\beta||_2^2 \leq \lambda^2$. Other steps are
the same as the $\ell_\infty$ case. For the convergence analysis of our LP-ADMM, the convergence result has already covered
the general norm setting. We will add it in the revision.

**Q2**: Test performance on smaller datasets, and demonstrate it worths solving $\kappa < \infty$, comparing to $\kappa = \infty$.

**A2**: The test performance on smaller datasets has already been done in the previous work, see Table 1 in the reference
[21]. Thank you for your suggestions. We will better motivate the results and clarify their relationship with the literature
in the revision.

**Q3**: What is the information-theoretic lower bound of the DRLR problem? This is related to the last sentence in the
conclusion section. Discuss whether the algorithm achieves the optimal complexity.

**A3**: Thank you for pointing out the interesting research direction. We mentioned it in the paper, see Remark 6.8 in the
appendix and reference [31]. We have proved that the $\beta$-subproblem (1.2) enjoys the Luo-Tseng error bound. Thus, the
optimal local convergence rate is linear for first-order algorithms theoretically. However, only a sublinear rate (i.e.,
$\mathcal{O}(\frac{1}{K})$)) has been established in our paper since it is open whether the primal-dual error bound holds for the problem
(1.3). Under the ADMM framework, to the best of our knowledge, this is the best complexity bound to date. However,
it remains open whether one can prove that our proposed LP-ADMM or some other algorithms can achieve the optimal
complexity (i.e., linear rate) for the $\beta$-subproblem.

**Q4**: The relationship between the KKT point of (2.1) and the global solution of the original problem? In particular, if
we get an eps-optimal solution of (2.1), what is the optimality gap of the original DRLR problem?

**A4**: Firstly, the KKT point of the problem (2.1) (i.e., $x^*$) is the corresponding optimal solution of the original problem.
Furthermore, if $||(x_k, y_k, w_k) - (x^*, y^*, w^*)|| \leq \epsilon$, then we have $||x_k - x^*|| \leq ||(x_k, y_k, w_k) - (x^*, y^*, w^*)|| \leq \epsilon$.
Then, we have the $\epsilon$-optimal solution of the original problem.

[Meta-Review · NeurIPS 2019]

The paper proposes to solve the Wasserstein distributionally robust logistic regression problem by using an ADMM-style algorithm. All reviewers and myself find it interesting and can have practical and theoretical impacts. Clear acceptance.